# A Comparison of Evans Blue and 4-(*p*-Iodophenyl)butyryl Albumin Binding Moieties on an Integrin α_v_β_6_ Binding Peptide

**DOI:** 10.3390/pharmaceutics14040745

**Published:** 2022-03-30

**Authors:** Ryan A. Davis, Sven H. Hausner, Rebecca Harris, Julie L. Sutcliffe

**Affiliations:** 1Department of Biomedical Engineering, University of California, Davis, CA 95616, USA; rydavis@ucdavis.edu; 2Department of Internal Medicine, Division of Hematology/Oncology, University of California, Davis, CA 95817, USA; shhausner@ucdavis.edu (S.H.H.); relharris@ucdavis.edu (R.H.); 3Center for Molecular and Genomic Imaging, University of California, Davis, CA 95616, USA

**Keywords:** albumin binding moieties, peptides, Evans blue, 4-(*p*-iodophenyl)butyric acid, integrin α_v_β_6_, integrin α_v_β_6_ binding peptide, improved pharmacokinetics, PET imaging

## Abstract

Serum albumin binding moieties (ABMs) such as the Evans blue (EB) dye fragment and the 4-(*p*-iodophenyl)butyryl (IP) have been used to improve the pharmacokinetic profile of many radiopharmaceuticals. The goal of this work was to directly compare these two ABMs when conjugated to an integrin α_v_β_6_ binding peptide (α_v_β_6_-BP); a peptide that is currently being used for positron emission tomography (PET) imaging in patients with metastatic cancer. The ABM-modified α_v_β_6_-BP peptides were synthesized with a 1,4,7,10-tetraazacyclododecane-1,4,7,10-tetracetic acid (DOTA) chelator for radiolabeling with copper-64 to yield [^64^Cu]Cu DOTA-EB-α_v_β_6_-BP ([^64^Cu]**1**) and [^64^Cu]Cu DOTA-IP-α_v_β_6_-BP ([^64^Cu]**2**). Both peptides were evaluated in vitro for serum albumin binding, serum stability, and cell binding and internalization in the paired engineered melanoma cells DX3puroβ6 (α_v_β_6_ +) and DX3puro (α_v_β_6_ −), and pancreatic BxPC-3 (α_v_β_6_ +) cells and in vivo in a BxPC-3 xenograft mouse model. Serum albumin binding for [^64^Cu]**1** and [^64^Cu]**2** was 53–63% and 42–44%, respectively, with good human serum stability (24 h: [^64^Cu]**1** 76%, [^64^Cu]**2** 90%). Selective α_v_β_6_ cell binding was observed for both [^64^Cu]**1** and [^64^Cu]**2** (α_v_β_6_ (+) cells: 30.3–55.8% and 48.5–60.2%, respectively, vs. α_v_β_6_ (−) cells <3.1% for both). In vivo BxPC-3 tumor uptake for both peptides at 4 h was 5.29 ± 0.59 and 7.60 ± 0.43% ID/g ([^64^Cu]**1** and [^64^Cu]**2**, respectively), and remained at 3.32 ± 0.46 and 4.91 ± 1.19% ID/g, respectively, at 72 h, representing a >3-fold improvement over the non-ABM parent peptide and thereby providing improved PET images. Comparing [^64^Cu]**1** and [^64^Cu]**2**, the IP-ABM-α_v_β_6_-BP [^64^Cu]**2** displayed higher serum stability, higher tumor accumulation, and lower kidney and liver accumulation, resulting in better tumor-to-organ ratios for high contrast visualization of the α_v_β_6_ (+) tumor by PET imaging.

## 1. Introduction

The use of biologically active molecules such as peptides and antibodies continues to increase for both diagnosis and therapy [1,2,3]. Peptides are attractive platforms for diagnostics due to their ability to achieve high target binding affinity and in part due to their small size which results in short biological half-life and rapid clearance from non-target tissues, producing good target-to-non-target contrast, low toxicity, and generally low or absent immunogenicity [1]. Synthetic advantages of peptides include simple preparation and easy, flexible functionalization or chemical modification to further improve affinity, stability, selectivity, and overall pharmacokinetic properties [1,4]. However, some of the properties that are desirable for a diagnostic agent can hamper the translation to a therapeutic, which relies on a prolonged circulation for high and persistent uptake in the targeted tissue. Too rapid clearance can render the therapeutic ineffective, and poor clearance from non-target tissue can lead to off-target toxicity. Thus, peptides typically require fine-tuning for therapeutic applications to balance circulation time and provide high target accumulation with sufficient clearance from non-target tissues [5,6,7].

Chemical modifications of peptides offer a route to improving these pharmacokinetic properties; this includes incorporation of polyethylene glycol (PEG; PEGylation), glycosylation, or the formation of protein conjugates (e.g., with serum albumin) [4,8,9,10,11]. PEGylation is a convenient approach as PEGs are commercially available in a variety of molecular sizes, including mono-disperse PEGs with various functional groups for synthetic orthogonality [1,9]. PEGylation increases hydrophilicity (reducing kidney, lung, and liver accumulation) [12,13], provides increased stability (by protection from proteases), and reduces immunogenicity (by masking the peptide) [9,13]. The size and placement of the PEG on the peptides can significantly affect the pharmacokinetics and tumor accumulation [12,13,14]. Stability, circulation time, and tumor uptake of peptides can also be increased by chemical ligation ex vivo to serum albumin (taking advantage of albumin’s size, long circulation time, and renal recycling) [8,15,16,17]. Alternatively, the same benefit can be achieved by direct attachment of a small albumin binding moiety (ABM) onto the peptide without substantially increasing the size. The ABM binds reversibly to albumin in the blood, thereby increasing circulation time and facilitating renal recycling, which, in turn, increases target tissue accumulation [8,15,16,18]. Several ABMs have been employed to modify pharmaceuticals currently used in the clinic, with some being used on their own, primarily for measuring plasma volume [16]; among the first ABMs used to modify pharmaceuticals were long-chain fatty acids, such as myristic and palmitic acid [5], and later other lipophilic molecules including benoxaprofen, phenytoin, ibuprofen, and naproxen [16].

More recently, two ABMs in particular, a fragment of Evans blue (EB) dye and the 4-(*p*-iodophenyl)butytryl (IP) group, have also been used to modify the pharmacokinetic profile of radiopharmaceuticals, in particular small molecules (folic acid and prostate specific membrane antigen (PSMA) agents) and peptides (octreotide, exendin-4, and cRGDfK) [10,11,16,18,19,20]. The EB-ABM was derived from Evans blue dye, a dye which has been used clinically for over 90 years to measure plasma volume and determine blood-brain barrier integrity [16,17,21]. The EB-ABM fragment was first used in 2004 as an MRI contrast agent for imaging blood vessels [22] and has since been used for a variety of applications, including determining blood volume, vascular permeability, and as a conjugate to enhance receptor targeting agents (small molecules and peptides) for both cancer imaging and therapy [9,17,23,24,25,26]. The IP-ABM has also been studied extensively to enhance radiopharmaceuticals (small molecules and peptides), where the group at the *para*-position of the aromatic ring of the IP-ABM can be tuned to adjust serum albumin affinity [15,27,28], and a neighboring aspartate residue (D) has been shown to provide a more sustained tumor retention [29]. Numerous preclinical studies have evaluated both ABMs and noted prolonged blood circulation, with an increase in tumor uptake that can also lead to a reduction of kidney accumulation [7,11,18].

The Sutcliffe laboratory has spent over a decade developing and optimizing an integrin α_v_β_6_-binding peptide (α_v_β_6_-BP) [30] to selectively target integrin α_v_β_6_, an epithelium cell surface receptor that is absent or expressed in low levels in healthy adult epithelia, but is highly expressed in numerous challenging cancers, where it is associated with angiogenesis, proliferation, invasion, metastasis, and chemoresistance [31,32,33,34,35,36,37,38,39,40,41]. Thus, the integrin α_v_β_6_ has been recognized as negative prognostic indicator with the expression levels correlating to poor prognosis and overall survival in many cancers [31,32,33,34,35,36,37,38,39,40,41]. During the optimization of the α_v_β_6_-BP, the bi-terminal PEGylation with monodispersed PEG_28_ of the 20 amino acid A20FMDV2-peptide (NAVPNLRGDLQVLAQKVART) derived from the integrin α_v_β_6_-targeting foot and mouth disease virus, showed greatly improved integrin α_v_β_6_ affinity and selectivity, and improved on the peptide’s stability and tumor accumulation and retention [14]. Since then, further modifications have been tested in numerous preclinical models with an advancement of the peptide to >10-fold increase in tumor accumulation and the successful translation of the 4-[^18^F]fluorobenzoyl labeled [^18^F]α_v_β_6_-BP into the clinic for PET imaging of a variety of cancers, including pancreatic adenocarcinoma [30]. Further optimization of α_v_β_6_-BP continues towards an integrin α_v_β_6_ targeted peptide receptor radionuclide therapy (PRRT).

Recently, Hausner et al. described the IP-ABM modified α_v_β_6_-BP radiolabeled using 1,4,7-triazacyclo-nonane-*N*,*N*’,*N*”-triacetic acid (NOTA) for aluminum [^18^F]fluoride chelation, with the goals of improving the biodistributions and simplifying the fluorine-18 radiochemistry [42]. The [^18^F]AlF NOTA-IP-ABM-α_v_β_6_-BP had increased blood circulation and tumor accumulation that allowed for high-contrast PET imaging at 6 h post-injection (p.i.) [42], and >3.5-fold lower kidney retention than the very early generation [^18^F]AlF NOTA-A20FMDV2-peptide [43]. Building on these data and to extend the imaging window beyond that of fluorine-18 (t_1/2_ = 109.7 min), a copper-64 1,4,7,10-tetraazacyclododecane-1,4,7,10-tetracetic acid (DOTA) IP-ABM-α_v_β_6_-BP (t_1/2_ = 12.7 h) was prepared, which again resulted in an increased tumor accumulation that allowed PET imaging up to 72 h p.i. [44].

In the present study, we describe a head-to-head comparison of the α_v_β_6_-BP modified with either EB-ABM or IP-ABM, with the goal to examine if fine tuning of the ABM could further increase tumor accumulation. Copper-64 radiolabeled [^64^Cu]Cu DOTA-EB-α_v_β_6_-BP ([^64^Cu]**1**) and [^64^Cu]Cu DOTA-IP-α_v_β_6_-BP ([^64^Cu]**2**), along with the non-α_v_β_6_-targeting ABM controls [^64^Cu]Cu DOTA-EB ([^64^Cu]**3**) and [^64^Cu]Cu DOTA-IP ([^64^Cu]**4**; Figure 1) were synthesized. Peptides [^64^Cu]**1** and [^64^Cu]**2** were evaluated in vitro by competitive ELISA, serum stability, albumin binding assays, and cell binding and internalization assays with DX3puroβ6 (α_v_β_6_+), DX3puro (α_v_β_6_−), and BxPC-3 (α_v_β_6_+) cells (against controls [^64^Cu]**3** and [^64^Cu]**4**), and in vivo by PET/CT imaging and biodistribution studies in mice bearing BxPC-3 xenograft tumors (4–72 h, p.i., against controls [^64^Cu]**3** and [^64^Cu]**4** at 4 h, p.i.).

## 2. Materials and Methods

### 2.1. Materials and General Information

Amino acids *N*-terminally protected with a fluorenylmethyloxycarbonyl (Fmoc) protecting group and acid labile side chain protecting groups (trityl, Pbf, *tert*-butyl, or Boc) were purchased from Novabiochem (MA, USA) or GL Biochem (Shanghai, China). The orthogonally protected lysine with a 1-(4,4-dimethyl-2,6-dioxocyclohex-1-ylidene)-3-methylbutyl (ivDde) sidechain protecting group and an *N*-terminal Fmoc protecting group, Fmoc-Lys(ivDde)-OH was purchased from ChemPep (Wellington, FL, USA) and the reverse ivDde-Lys(Fmoc)-OH was purchased from EMD (MA, USA). The Fmoc-NH-PEG_28_ carboxylic acid was purchased from Polypure (Oslo, Norway) and the chelator DOTA-tris(*tert*-butyl ester) was purchased from CheMatech (Dijon, France) and Macrocyclics (Plano, TX, USA). The coupling reagent 1-[bis(dimethylamino)methylene]-1*H*-1,2,3-triazolo[4,5-b]pyridinium 3-oxid hexafluorophosphate (HATU) was purchased from GL Biochem, and benzotriazol-1-yl-oxytripyrrolidinophosphoniumhexafluorophosphate (PyBOP) was purchased from Novabiochem. Ethylenediaminetetraacetic acid (EDTA), manganese chloride (MnCl_2_), and Tris were purchased from Sigma-Aldrich (St. Louis, MO, USA). Tween 20 and sodium chloride (NaCl) were purchased from Fisher (Hampton, NH, USA). The non-fat dry milk powder was purchased from Raley’s (West Sacramento, CA, USA). Anhydrous *N*,*N*-diisopropylethylamine (DIPEA) and hydrazine were purchased from Sigma-Aldrich and used without additional purification. Solvents *N*,*N*-dimethylformamide (DMF), dimethylsulfoxide (DMSO), acetonitrile (ACN), methanol (MeOH), dichloromethane (DCM), ethyl acetate (EtOAc), *n*-hexanes, and pyridine were purchased from EMD or Acros (NJ, USA). Water used was purified with a Millipore Integral 5 Milli-Q water system at 18.2 MΩ/cm resistivity through a 0.22 μm filter. All solid phase couplings were carried out by rotation in a fritted polypropylene reactor. Thin-layered chromatography (TLC) plates (silica gel 60 with 254 nm fluorescent indicator) from EMD were visualized by UV lamp at 254 nm and/or iodine staining (for the synthesis of **6**). Purification of compound **6** was carried out by normal phase flash column chromatography with silica gel (40–63 μm; Silicycle, QC, Canada). Characterization, purity, and stability were assessed by analytical C_12_-reverse-phase (RP) high-pressure liquid chromatography (HPLC) column (Jupiter Proteo, 250 mm × 4.6 mm × 4 μm; Phenomenex, Torrance, CA, USA). A Semi-preparative C_18_-RP-column (Proteo-Jupiter, 250 mm × 10 mm × 10 μm; Phenomenex) was used for purification as described in the Appendix A. All RP-HPLC were carried out on a Dionex Ultimate 3000 HPLC system or a Beckman Coulter Gold HPLC with the latter being used for all radio-RP-HPLC analysis. RP-HPLC were monitored by a UV detector at a wavelength of 220 nm; a serially connected gamma detector was used to monitor radioactivity. [^64^Cu]CuCl_2_ was from the University of Wisconsin Medical Physics Department (WIMR Cyclotron Labs, Madison, WI, USA). Tissue culture and cellular assays used Dulbecco’s Modified Eagle Medium (DMEM), Roswell Park Memorial Institute (RPMI) 1640 medium, fetal bovine serum (FBS), bovine serum albumin (BSA), penicillin-streptomycin-glutamine (PSG), puromycin, and phosphate buffered saline (PBS; all: Gibco/Thermo Fisher). The DX3puroβ6 and DX3puro cells were a gift from Dr. John Marshall. The DX3puroβ6 and DX3puro cell lines were maintained in DMEM medium, supplemented with 10% FBS, 1% penicillin-streptomycin-glutamine, and 2 mg/mL-puromycin. The BxPC-3 cells were purchased from American Type Culture collection (ATCC, Manassas, VA, USA) and maintained in RPMI 1640 medium supplemented with 10% FBS and 1% penicillin-streptomycin-glutamine. Cells were kept in a humidified incubator at 37 °C under a 5%-carbon dioxide atmosphere. A Wizard 1470 or Wizard^2^ 2470 automatic γ-counter (Perkin-Elmer, Waltham, MA, USA) was used to measure radioactivity samples. Mass spectrometry analysis was performed at the UC Davis Mass Spectrometry Facility using either a matrix assisted laser desorption ionization time of flight (MALDI- TOF) spectrometer (UltraFlextreme; Bruker, Billerica, MA, USA) in positive ionization mode with a sinapic acid matrix (Sigma-Aldrich), or with electrospray ionization (ESI) using a quadrupole ion-trap mass spectrometer (Orbitrap; ThermoFisher). Nuclear magnetic resonance (NMR) spectra were collected at the UC Davis NMR Facility on an 800 MHz Bruker instrument with the chemical shifts referenced to the residual solvent of deuterium oxide (D_2_O, HOD 4.79 ppm).

### 2.2. Synthesis of EB-ABM 8

The synthesis of the Evans blue fragment (EB-ABM **8**) was based on previously described methods [7,17,45] (Figure 1). In brief, *o*-tolidine **5** (531 mg, 2.5 mmol; TCI America, Inc., OR, USA) was dissolved in anhydrous pyridine (1 mL) followed by the addition of succinic anhydride (300 mg, 3.0 mmol) in DMF (1 mL) and allowed to react overnight at room temperature. The crude reaction mixture was concentrated under vacuum and purified by silica-gel column chromatography using a four solvent gradient system beginning with EtOAc/n-hexanes (1/1, *v*/*v*) to remove unreacted *o*-tolidine (**5**, yellow band). The solvent was then changed to 100% EtOAc before switching to MeOH/DCM (1/9, *v*/*v*) and gradually ramping to 3/7 (*v*/*v*) to obtain **6** (648 mg, rt = 0.13, 1:1 hexanes:EtOAc) as a white solid in 83% yield. Compound 6 was analyzed by analytical RP-HPLC and ESI mass spectrometry (Appendix A).

Compound **6** (300 mg, 0.96 mmol) was added to a 25 mL round bottom flask with stir bar containing MeOH (7 mL) and water (5 mL). The contents were cooled to 0 °C (ice/brine solution) and allowed to stir for 15 min prior to addition of concentrated hydrochloric acid 240 μL (HCl, 12.1 N; EMD). The diazonium formation of **7** was most successful when the addition of sodium nitrite was done in two portions; the first portion of sodium nitrite (NaNO_2_, 70 mg, 1.01 mmol; Sigma-Aldrich) was allowed to react for 5 min before the addition of the second portion (NaNO_2_, 70 mg, 1.01 mmol), after which the reaction was stirred an additional 30 min to generate **7** in situ, which was produced in better yields using the methanol co-solvent than water alone [46]. During in situ formation of **7**, sodium bicarbonate (350 mg, 4.17 mmol; EMD) was dissolved in water (4 mL) with 1-amino-8-napthol-2,4-disulfonic acid (377 mg, 1.18 mmol; TCI America, Inc.) in a separate 25 mL round bottom flask and the contents cooled in an ice/brine solution (~20 min). Next, the diazonium **7** reaction mixture (yellow) was cannulated into the 1-amino-8-napthol-2,4-disulfonic acid (brown-purple) solution by drop-wise addition over 20 min while maintaining both solutions at 0 °C. Upon complete addition of **7**, the reaction contents were allowed to stir for 3 h at 0 °C, and the crude reaction mixture was lyophilized and purified by semi-preparative RP-HPLC, and the collected material lyophilized. The EB-ABM **8** was afforded as a fluffy purple solid (480 mg, 78%) and was analyzed by analytical RP-HPLC, ESI mass spectrometry, and NMR (Appendix A).

### 2.3. Synthesis of DOTA-ABM-α_v_β_6_-BPs ***1*** and ***2***

The α_v_β_6_-BP (PEG_28_-NAVPNLRGDLQVLAQRVART-PEG_28_) was synthesized on NovaSyn TGR resin (NovaBiochem) and PEGylation was done using monodisperse Fmoc-amino-PEG-propionic acid (Fmoc-PEG_28_-CO_2_H; FW = 1544.8 g/mol) as previously described [30] using standard Fmoc-chemistry. After each coupling or deprotection the resin was rinsed with DMF (3×), MeOH (3×), and DMF (3×). The α_v_β_6_-BP-resin was split in equal portions (100 mg, 0.0088 mmol) and further modified at the *N*-terminus for the synthesis of peptides **1** and **2**. The DOTA-EB-α_v_β_6_-BP **1** was generated by first removing the *N*-terminal Fmoc of the α_v_β_6_-BP with 20% piperidine (Sigma-Aldrich) in DMF (2 × 10 min) followed by the addition of Fmoc-Lys(ivDde)-OH (50.6 mg, 0.088 mmol) using HATU (32.3 mg, 0.085 mmol) and DIPEA (30 μL, 0.172 mmol) in DMF (1 mL) for 2 h. The Fmoc was subsequently removed with 20% piperidine in DMF (2 × 10 min) and DOTA-tris(*tert*-butyl ester) (50.3 mg, 0.088 mmol) was coupled for 2 h to the *N*-terminus with HATU (32.3 mg, 0.085 mmol) and DIPEA (30 μL, 0.172 mmol) in DMF (1 mL). The removal of the ivDde lysine-sidechain protecting group was done with hydrazine (50 μL) in DMF (1 mL, 2 × 30 min) and the resin dried under vacuum. The EB-ABM **8** (60 mg, 0.093 mmol) was then coupled to the ε-amine of the sidechain of the DOTA-lysine on the α_v_β_6_-BP-resin using PyBOP (125 mg, 0.24 mmol) and DIPEA (50 μL, 0.287 mmol) for 6 h to yield DOTA-EB-α_v_β_6_-BP-resin **1** (Appendix A). The DOTA-EB-α_v_β_6_-BP **1** was cleaved off the resin with concomitant removal of the protecting groups using trifluoroacetic acid (TFA, 2 mL; EMD), triisopropylsilane (TIPS, 50 μL; Alfa Aesar, Haverhill, MA, USA) and water (50 μL), concentrated, purified, and characterized by analytical RP-HPLC and MALDI-TOF (Appendix A). The IP-ABM containing-α_v_β_6_-BP **2** was prepared as previously described [42,44] were upon removal the *N*-terminal Fmoc of the α_v_β_6_-BP-resin, a ivDde-Lys(Fmoc)-OH was coupled. Completion of DOTA-IP-α_v_β_6_-BP **2** was done by sequential coupling/deFmocing of (1) Fmoc-Asp(OtBu)-OH, (2) *N*-γ-Fmoc-γ-aminobutyric acid, and (3) 4-(*p*-iodophenyl)butyric acid using HATU and DIPEA for each coupling. Completion of DOTA-IP-α_v_β_6_-BP **2** was achieved by removal of the *N*-terminal ivDde protecting group with 5%-hydrazine in DMF followed by attachment of DOTA-tris(*tert*-butyl ester) [44]. The completed DOTA-IP-α_v_β_6_-BP **2** was cleaved, purified, and analyzed as described above for the DOTA-EB-α_v_β_6_-BP **1** (Appendix A) [44].

### 2.4. Synthesis of Non-Targeting ABMs ***3*** and ***4***

Using Fmoc-chemistry with Rink AM resin (200 mg, 0.114 mmol; GL Biochem), DOTA-ABM non-targeting compounds **3** and **4** were synthesized by first coupling Fmoc-Lys(ivDde)-OH (196.6 mg, 0.342 mmol) using HATU (123.5 mg, 0.325 mmol) and DIPEA (100 μL, 0.574 mmol) in DMF (1 mL). The Fmoc was removed with 20% (*v*/*v*) piperidine in DMF (1 mL, 2 × 10 min) and DOTA-tris(*tert*-butyl ester) (80 mg, 0.140 mmol) was coupled for 2 h with HATU (50 mg, 0.132 mmol) and DIPEA (50 μL, 0.287 mmol). Following the ivDde protecting group removal with hydrazine (50 μL) in DMF (1 mL, 2 × 30 min), the resin was dried under vacuum and split into equal portions for synthesis of **3** and **4**. For **3**, the EB-ABM **8** (165 mg, 0.256 mmol) was coupled using PyBOP (166.5 mg, 0.32 mmol) and DIPEA (100 μL, 0.574 mmol) for 6 h. The EB-ABM **3** was cleaved off the resin, purified, and characterized by analytical RP-HPLC and MALDI-TOF (Appendix A). IP-ABM **4** was prepared by sequential coupling/deFmocing of (1) Fmoc-Asp(OtBu)-OH (90 mg, 0.219 mmol), (2) *N*-γ-Fmoc-γ-aminobutyric acid (70 mg, 0.215 mmol), and (3) 4-(*p*-iodophenyl)butyric acid (65 mg, 0.224 mmol) using HATU (78 mg, 0.205 mmol) and DIPEA (100 μL, 0.574 mmol) for each coupling. The IP-ABM **4** was then cleaved off the resin, purified, and characterized by analytical RP-HPLC and MALDI-TOF (Appendix A).

### 2.5. Radiochemical Synthesis of [^64^Cu]***1***–***4***

DOTA-compounds **1**–**4** were dissolved in metal free water at 1 μg/μL, and the [^64^Cu]CuCl_2_ (1–10 μL of 0.5 M HCl, **1** and **2**: 174–255.3 MBq, **3** and **4**: 51–55 MBq) was diluted with 1.0 M ammonium acetate (NH_4_OAc, Sigma-Aldrich) aqueous solution (pH = 8.0) to 0.27 μL/MBq. Peptides **1** and **2** were added to the NH_4_OAc buffered [^64^Cu]CuCl_2_ such that the starting molar activity of the reaction was between 18.5 and 20 GBq/μmol. The starting molar activity for compounds **3** and **4** was between 15.9 and 17.1 GBq/μmol. The reaction mixtures were vortexed and warmed to 37 °C for 30 min. The radiochemical purity was assessed by quenching an aliquot of the reaction (≤1 μL; 0.74–3.7 MBq) with 0.1 M EDTA (50 μL) and analyzed by analytical RP-HPLC. Product identity was confirmed by cold spike RP-HPLC, i.e., co-injection of the radiolabeled product with authenticated respective [^Nat^Cu]Cu **1**–**4** reference standard of each compound (Appendix A). [^Nat^Cu]Cu **1**–**4** reference standards were produced via reaction of DOTA-compounds **1**–**4** (0.1–0.5 mg) with excess CuCl_2_ (Sigma-Aldrich, 1–6 mg) in water (50 μL) for 30 min at room temperature, and purified directly by RP-HPLC and confirmed by MALDI-TOF (Appendix A).

### 2.6. Integrin α_v_β_6_ Affinity ELISA

Affinity for the integrin α_v_β_6_ was determined by competitive binding ELISA of [^Nat^Cu]**1** and [^Nat^Cu]**2** against biotinylated-LAP (G&P Biosciences, Santa Clara, CA, USA) as previously described to determine the half-maximum inhibitory concentration (IC_50_) [44]. Briefly, in a 96 well Nunc Immuno maxisorp plate, capturing anti-α_v_ antibody (P2W7, 5 μg/mL, Abcam, MA, USA) was plated (50 μL/well) at 37 °C for 1 h, washed with PBS (3×), and blocked overnight with blocking buffer (300 μL/well, 0.5% non-fat dry milk powder (*w*/*v*), 1% Tween 20, in PBS). It was then washed with wash buffer that consisted of 2 mmol/L of Tris buffer (pH = 7.6), 150 mmol/L sodium chloride, 1 mmol/L manganese chloride, and 0.1% Tween 20 (*v*/*v*) in deionized water (3×). Purified integrin α_v_β_6_ (R&D Systems, Minneapolis, MN, USA) in conjugate buffer (50 μL/well, 20mM Tris, 1 mM MnCl_2_, 150 mM NaCl, 0.1% Tween, 0.1% milk powder in water) was then added to each well, incubated at 37 °C for 1 h, followed by washing using wash buffer 3×). Serial dilutions of each peptide stock of 2 mmol/L in 10% DMSO (*v*/*v*) into PBS and biotinylated natural ligand LAP were premixed in equal volumes and placed onto the plate in triplicate for each peptide concentration (50 μL/well) and allowed to incubate at 37 °C for 1 h then washed with wash buffer (3×). A 1:1000 dilution of ExtrAvidin Horseradish Peroxidase (HRP; Fisher, NH, USA) was added to each well (50 μL/well), incubated at 37 °C for 1 h, and then washed with wash buffer (3×). The ExtrAvidin HRP was detected with TMB One solution (50 μL/well; Promega Corp., Madison, WI, USA) for 10–15 min at room temperature. The reaction was stopped by adding 1N sulfuric acid (H_2_SO_4_, 50 μL/well; EMD, MA, USA) and the absorbance was measured in a Multiscan Ascent plate reader (Thermo Fisher, Waltham, MA, USA) at 450 nm. Half-maximal inhibitory concentration (IC_50_) of peptides was determined by fitting to sigmoidal dose-response model in GraphPad Prism 8.0 (GraphPad, CA, USA). For the positive control no peptide was added and for the negative controls either no biotinylated-LAP or no integrin α_v_β_6_ was added.

### 2.7. Cell Binding and Internalization Assay

Binding of [^64^Cu]**1**–**4** and internalization to DX3puro, DX3puroβ6, and BxPC-3 cells were determined as previously described [44]. Prior to the experiment, the cells were analyzed by flow cytometry to confirm levels of integrin α_v_β_6_ expression. Non-fat dry milk powder (0.5% *w*/*v* in PBS) was used to pretreat the assay tubes to prevent non-specific binding. Aliquots of [^64^Cu]**1**–**4** (≤1 μL, 7.4–18.5 KBq) in 50 μL serum free medium (pH 7.2) were added to a cell suspension (3.75 × 10^6^ cells in 50 μL serum free medium) and incubated for 1 h at room temperature in closed microfuge tubes (*n* = 3/cell line/compound) and gently agitated every 3 min to ensure mixing. The cells were pelleted by centrifugation at 200 (RCF) for 3 min and the supernatant collected. The cell pellet was washed with 0.5 mL serum free medium and the wash medium combined with the original supernatant. The cells were resuspended in 0.6 mL serum free medium for γ-counting. The fraction of bound radioactivity was determined with a γ-counter (by measuring cell pellet and combined supernatants). To determine the fraction of internalized radioactivity, the cells were re-pelleted, and re-suspended in acidic wash buffer (0.2 mol/L sodium acetate, 0.5 mol/L sodium chloride, pH 2.5, 300 μL, 4 °C, 5 min) to release surface-bound activity, followed by a wash with PBS (300 μL). The internalized fraction was determined with a γ-counter (cell pellet vs. radioactivity released into supernatant).

### 2.8. Human and Mouse Serum Binding Assay and Stability Assay

Serum protein binding of [^64^Cu]**1** and [^64^Cu]**2** was assessed following the previously reported method [42]. Peptides [^64^Cu]**1** and [^64^Cu]**2** were evaluated by ultrafiltration using Centrifree Ultrafiltration devices (EMD) according to the manufacturer’s recommendations. Experiments were carried out in triplicate. The Centrifree Ultrafiltration devices were pretreated with PBS containing Tween 20 (5% *v*/*v*), followed by triplicate rinses with PBS. An aliquot of each peptide [^64^Cu]**1** or [^64^Cu]**2** in PBS (≤25 μL, 20–60 KBq) was thoroughly mixed with 0.5 mL of serum at 37 °C in a microfuge tube. The mixture was incubated at 37 °C for 5 min, and an aliquot (50 μL) was transferred to a tube for γ-counting. The remaining sample was transferred to a Centrifree Ultrafiltration device and centrifuged for 40 min at 1500 (RCF) at ambient temperature (20–24 °C). An aliquot (50 μL) of the filtrate was transferred to a tube for γ-counting. For each radiolabeled peptide, a blank was run using 0.5 mL PBS/Tween 20 (5% *v*/*v*) instead of serum (*n* = 3) to determine non-specific binding. Following γ-counting, the protein-bound radioactivity was calculated by subtracting the counts measured in the filtrate aliquot (i.e., not protein-bound) from the counts in the corresponding serum aliquot. The data are expressed as mean ± standard deviation of fraction of radioactivity bound to protein after subtraction of non-specific binding determined in the blank.

For serum stability, mouse serum or human serum (0.5 mL, both purchased from Sigma-Aldrich) was combined with an aliquot of each of the peptides [^64^Cu]**1** and [^64^Cu]**2** (≤25 μL, 14.8–22.2 MBq) and incubated at 37 °C. At each time point (1, 4, and 24 h) an aliquot (50–200 μL) was taken, proteins precipitated with ethanol, and removed by pelleting at 1500 (RCF) for 4 min. The ethanol solution was diluted with water (1 mL) and analyzed by RP-HPLC as previously described [47].

### 2.9. Biodistribution

All animal procedures conformed to the Animal Welfare Act and were approved by the University of California, Davis Institutional Animal Care and Use Committee. Female athymic nu/nu-nude mice (6–8 weeks old) were purchased from Charles River Laboratories (Wilmington, MA, USA) and provided food and water on an ad libitum basis. BxPC-3 xenografts were implanted according to previous methods [42,44]. Briefly, BxPC-3 cells were evaluated by flow cytometry to confirm integrin α_v_β_6_ expression levels, injected subcutaneously into the left flank [5 million in 100 μL of a 1:1 mixture of serum-free RPMI and GFR Matrigel (Corning, New York, NY, USA)], and allowed to grow for approximately 3 weeks until tumors reached a diameter of 0.5–1 cm.

For biodistribution studies the [^64^Cu]**1**–**4** (3.7–5.55 MBq) in PBS (100 μL, pH 7.2) was injected intravenously (i.v.) via catheter into the tail vein. Following a conscious uptake period, the mice were anesthetized (5% isoflurane), euthanized, and dissected ([^64^Cu]**1** and [^64^Cu]**2**, *n* = 3/radiolabeled peptide/time point; 4, 24, and 48 h p.i.; the 72 h time point was obtained from the imaging animals after the 72 h PET/CT scans; compounds [^64^Cu]**3** and [^64^Cu]**4,**
*n* = 2/radiolabeled compound at 4 h p.i.). Tissues were rapidly collected, weighed, and radioactivity measured with a γ-counter. Decay-corrected radioactivity concentrations are expressed as the percentage of injected dose per gram of tissue (% ID/g). Data are reported as mean ± standard deviation (SD) (Appendix A).

### 2.10. Blocking Biodistribution

For blocking studies, the metal free peptides **1** or **2** (~220 nmol, 1.3 mg in 100 μL PBS), respectively, were injected i.v. (*n* = 1/peptide) as described above 10 min prior to the injection of matching radiolabeled [^64^Cu]**1** or [^64^Cu]**2** (3.7–5.55 MBq, 100 μL PBS). After a conscious 4 h uptake period, the animals were anesthetized, sacrificed, tissues rapidly collected, and analyzed as described above. Decay-corrected radioactivity concentrations are expressed as a percentage of injected dose per gram of tissue (% ID/g) (Appendix A).

### 2.11. PET-Imaging

For imaging studies, [^64^Cu]**1** and [^64^Cu]**2** (7.77–8.88 MBq) in PBS (100 μL, pH 7.2) were injected i.v. via a catheter into the tail vein of mice (*n* = 3/radiolabeled peptide) anesthetized with 2–3% isoflurane in medical grade oxygen. Animals were imaged in a prone position two at a time side by side. PET/CT scans were acquired using Inveon scanners (Inveon DPET scanner and Inveon SPECT/CT scanner, Siemens Medical Solutions, Knoxville, TN, USA; PET scans: a static 15 min scan at 4 h p.i., static 30 min scans at 24 and 48 h p.i., and a static 1 h scan at 72 h p.i.) and analyzed as previously described using the Inveon Research Workplace software (Siemens) [42,44].

### 2.12. Statistical Analysis

Quantitative data are reported as mean ± standard deviation (SD). Statistical significance was determined by a paired two-tailed Student’s *t*-test from the two independent sample means to give a significance value (*p*-value) at 95% confidence interval (CI). A *p*-value of <0.05 was considered statistically significant.

## 3. Results

### 3.1. Synthesis of EB-ABM 8

EB-ABM **8** was generated efficiently in three synthetic steps from *o*-tolidine **5** in an overall yield of 65% (Figure 1A). EB-ABM **8** was characterized by analytical RP-HPLC with a retention time of 17.72 min; ESI-MS m/z [M + H]^+^ for C_28_H_27_N_4_O_10_S_2_ calc’ed 643.1163; found 643.1207, and by ^1^H NMR (Appendix A). ^1^H NMR (800 MHz, D_2_O) δ 8.28 (s, 1H), 7.55 (d, *J* = 9.4 Hz, 1H), 7.29−7.27 (m, 2H), 7.24–7.23 (m, 1H), 7.17–7.16 (m, 1H), 7.13–7.12 (m, 1H), 6.98–6.97 (m, 1H), 6.88 (d, 7.8 Hz, 1H), 2.64–2.61 (m, 4H), 2.12 (s, 3H), 1.96 (s, 3H).

### 3.2. Synthesis and Radiochemical Synthesis of [^64^Cu]***1***–***4***

DOTA-compounds **1**–**4** were prepared in >97% isolated purity after RP-HPLC purification. DOTA-EB-α_v_β_6_-BP **1** had an RP-HPLC retention time of 17.22 min with a MALDI-TOF m/z [M + Na]^+^ for C_261_H_460_N_46_NaO_102_S_2_ calc’ed 5958.1556; found 5958.1756 (Appendix A). DOTA-IP-α_v_β_6_-BP **2** had an RP-HPLC retention time of 17.07 min with a MALDI-TOF m/z [M + H]^+^ for C_251_H_458_IN_44_O_98_ calc’ed 5786.1314; found 5786.1209 (Appendix A). DOTA-EB-ABM **3** had an RP-HPLC retention time of 14.18 min with a MALDI-TOF m/z [M + H]^+^ for C_50_H_66_N_11_O_17_S_2_ calc’ed 1156.4074; found 1156.4079 (Appendix A). DOTA-IP-ABM **4** had an RP-HPLC retention time of 14.46 min with a MALDI-TOF m/z [M + H]^+^ for C_40_H_63_IN_9_O_13_ calc’ed 1004.3585; found 1004.3590 (Appendix A).

The ^64^Cu-radiolabeled compounds ([^64^Cu]**1**–**4**) were produced in near quantitative yields (*n* = 2–4/compound/molar activity ranging between 16 and 20 GBq/μmol) by reaction of with [^64^Cu]CuCl_2_ in 1.0 M NH_4_OAc-buffer (pH = 8) at 37 °C for 30 min (Figure 1B). The radiochemical purities were ≥97% as determined analytical radio-RP-HPLC and compounds [^64^Cu]**1**–**4** used without further purification. Analytical radio-RP-HPLC retention times were: [^64^Cu]**1**—19.05 min (Appendix A); [^64^Cu]**2**—18.68 min (Appendix A); [^64^Cu]**3**—17.01 min (Appendix A); and [^64^Cu]**4**—16.73 min (Appendix A).

### 3.3. Integrin α_v_β_6_ Affinity ELISA

Competitive ELISA against biotinylated LAP, demonstrated that both ABM modifications of α_v_β_6_-BP were well tolerated; [^Nat^Cu]**1** and [^Nat^Cu]**2** showed high integrin α_v_β_6_-affinity as expressed by the half-maximum inhibitory concentrations (IC_50_); [^Nat^Cu]**1** and [^Nat^Cu]**2**: IC_50_ = 14 ± 2 and 19 ± 5 nM, respectively) compared to DOTA-α_v_β_6_-BP (IC_50_ = 28 ± 3 nM) [44].

### 3.4. Cell Binding and Internalization Assay

Cell binding studies showed that [^64^Cu]**1** and [^64^Cu]**2** both bound to cells in an α_v_β_6_-dependent manner at similar levels (DX3puroβ6 (+): [^64^Cu]**1** 55.8 ± 3.0% of total radioactivity, [^64^Cu]**2** 60.2 ± 3.9%; BxPC-3 (+): [^64^Cu]1 30.3 ± 2.7%, [^64^Cu]2 48.5 ± 3.5%; and the negative control DX3puro (−): [^64^Cu]**1** 2.7 ± 0.5%, [^64^Cu]**2** 3.1 ± 0.3%, Figure 2). This resulted in binding ratios for DX3puroβ6 (+) vs. DX3puro (−) of 20.7:1 for [^64^Cu]**1** and 19.4:1 for [^64^Cu]**2**. Internalization into α_v_β_6_-positive cells was also high ([^64^Cu]**1**: 48.5–52.7% of the bound radioactivity, [^64^Cu]**2**: 41.5–54.8%, Figure 2). The non-targeting control ABM conjugates [^64^Cu]**3** and [^64^Cu]**4** exhibited low, non-specific binding to all cell lines (≤4.3%; Appendix A).

### 3.5. Human and Mouse Serum Binding Assay and Stability Assay

Serum albumin binding for [^64^Cu]**1** and [^64^Cu]**2** was similar, with higher binding to human serum protein (53.4 ± 0.9% and 63.3 ± 1.5%, respectively) than to mouse serum protein (41.9 ± 1.1% and 44.0 ± 0.1%, respectively; Figure 3A). The ABM modifications of [^64^Cu]**1** and [^64^Cu]**2** increased the serum albumin affinity as the [^64^Cu]Cu DOTA-α_v_β_6_-BP without an ABM modification showed <29% binding to either serum albumin [44]. Both peptides showed high stability in human serum at 37 °C ([^64^Cu]**1** 1 h: 99% and 4 h: 89% intact; [^64^Cu]**2** 1 h: 99% and 4 h: 93% intact) with some degradation apparent after 24 h ([^64^Cu]**1**: 76% intact vs. [^64^Cu]**2**: 90% intact, Figure 3B). In contrast, faster degradation was observed in mouse serum at 37 °C, and the stability was lower for [^64^Cu]**1** than for [^64^Cu]**2** at all-time points; [^64^Cu]**1** was 78% intact at 1 h, dropping to 58% at 4 h, and largely metabolized at 24 h (14% intact). By comparison, [^64^Cu]**2** was 92% and 83% intact at 1 h and 4 h, respectively, with 48% remaining intact at 24 h, a 3.4-fold higher stability than [^64^Cu]**1** (Figure 3C).

### 3.6. Biodistribution

The biodistributions for [^64^Cu]**1** and [^64^Cu]**2** in the BxPC-3 tumor model showed good tumor uptake (4 h to 72 h: [^64^Cu]**1** 5.29 ± 0.59 to 3.32 ± 0.46% ID/g, [^64^Cu]**2** 7.60 ± 0.43 to 4.91 ± 1.19% ID/g, Figure 4). Overall, tumor uptake of [^64^Cu]**2** appeared higher than of [^64^Cu]**1**, particularly at the earliest time point, and relative tumor washout over the total observed time frame was similar for both peptides. The ABM modifications increased tumor accumulation by >3-to-4.5-fold compared to the [^64^Cu]Cu DOTA-α_v_β_6_-BP without an ABM, which had only 1.61 ± 0.70% ID/g at 4 h in the same BxPC-3 tumor model [44]. Clearance for [^64^Cu]**1** and [^64^Cu]**2** was primarily renal and the kidneys were the organ with the highest levels of radioactivity (Appendix A). Notably, [^64^Cu]**1** showed more than double the kidney uptake of [^64^Cu]**2** at 4 h, p.i. ([^64^Cu]**1** 75.51 ± 7.26% ID/g; [^64^Cu]**2** 33.56 ± 5.39% ID/g; *p* = 0.0013) and remained significantly higher for at least 48 h (>1.7-fold, *p* < 0.05), but both were cleared from the kidneys over time with accumulation dropping at 72 h ([^64^Cu]**1** 19.97 ± 6.91% ID/g; [^64^Cu]**2** 11.48 ± 1.02% ID/g; *p* = 0.103, Figure 4). Kidney accumulation for the ABM containing peptides [^64^Cu]**1** and [^64^Cu]**2** was initially higher than for the parent non-ABM containing [^64^Cu]Cu DOTA-α_v_β_6_-BP (20.37 ± 1.67% ID/g at 4 h to 6.81 ± 1.36% ID/g at 48 h) [44]. Some clearance for [^64^Cu]**1** and [^64^Cu]**2** was also observed through the gastrointestinal tract (GI), with the stomach having the highest uptake at 4 h, p.i. ([^64^Cu]**1**: stomach 6.41 ± 0.64% ID/g, small intestines 4.72 ± 0.55% ID/g, large intestines 4.13 ± 0.10% ID/g, [^64^Cu]**2**: stomach 18.07 ± 2.91% ID/g, small intestines 9.55 ± 1.21% ID/g, large intestines 9.83 ± 0.69% ID/g; Figure 4). Clearance from the GI tract was further confirmed by radioactivity measurements of fecal matter (4–72 h: [^64^Cu]**1**: 3.03 ± 0.67 to 1.81 ± 0.74% ID/g; [^64^Cu]**2**: 9.32 ± 1.08 to 2.29 ± 0.53% ID/g; Figure 4). The GI uptake for [^64^Cu]**2** was more than double that of [^64^Cu]**1** at the earliest time point, but both peptides dropped over time to below 3.2% ID/g at 72 h. The liver uptake was moderate (<3% ID/g) throughout for both peptides; but it increased to significantly higher levels for the EB-ABM containing peptide [^64^Cu]**1,** beginning at 24 h, reaching >1.8-fold higher levels than [^64^Cu]**2** at 72 h (2.36 ± 0.51% ID/g vs. 1.30 ± 0.13% ID/g, respectively; *p* = 0.025, Figure 4). Overall, the EB-ABM containing peptide [^64^Cu]**1** had a less favorable pharmacokinetic profile with significantly higher uptake in the kidneys and liver, resulting in generally lower tumor-to-tissue ratios for [^64^Cu]**1** compared to [^64^Cu]**2**, most notably for the tumor-to-kidney ratio ([^64^Cu]**1** 0.13 ± 0.06/1 to 0.19 ± 0.08/1 vs. [^64^Cu]**2** 0.20 ± 0.06/1 to 0.44 ± 0.14/1), and the tumor-to-liver ratio ([^64^Cu]**1** 2.39 ± 0.59/1 to 1.47 ± 0.47/1 vs. [^64^Cu]**2** 2.72 ± 0.62/1 to 3.77 ± 0.72/1) (Appendix A).

The non-α_v_β_6_-targeting ABM controls [^64^Cu]**3** and [^64^Cu]**4** were used to determine non-specific uptake and provide support that the enhanced tumor accumulation of ABM containing peptides [^64^Cu]**1** and [^64^Cu]**2** was due to integrin α_v_β_6_ receptor mediated uptake. The biodistributions of the non-α_v_β_6_-targeting ABM controls [^64^Cu]**3** and [^64^Cu]**4** at 4 h p.i. (*n* = 2/compound) showed prolonged blood circulation with much higher blood accumulation (38.9 ± 10.4% ID/g and 9.5 ± 1.3% ID/g, respectively; Figure 5, Appendix A). This increased blood accumulation also led to higher systemic accumulation in other tissues, especially the highly perfused tissues such as the heart, muscle, liver, and lung (Figure 5, Appendix A), with the exception of the kidneys (18.6 ± 1.4% ID/g and 4.34 ± 0.61% ID/g, respectively). These distinctly different pharmacokinetic profiles of the non-integrin α_v_β_6_-targeting [^64^Cu]**3** and [^64^Cu]**4** resulted in a low tumor-to-blood ratio of <0.9/1 compared to >4/1 for [^64^Cu]**1** and [^64^Cu]**2**, a lower tumor-to-muscle ratio ranging from 5.6 to 6.3/1 for [^64^Cu]**3** and [^64^Cu]**4** compared to >8/1 for [^64^Cu]**1** and [^64^Cu]**2**, and a lower tumor-to-liver ratio of 1.2–1.3/1 for [^64^Cu]**3** and [^64^Cu]**4** compared to 3.2–4.9/1 for [^64^Cu]**1** and [^64^Cu]**2** (Figure 5, Appendix A).

### 3.7. Blocking Biodistribution

Integrin α_v_β_6_-dependence of the tumor uptake was further substantiated by blocking studies with pre-administration of the respective nonradioactive peptide, which reduced tumor uptake to 2.91% ID/g and 2.89% ID/g for [^64^Cu]**1** and [^64^Cu]**2**, respectively (4 h; Δ = −45% and −62%, *p* = 0.0124 and 0.0007, respectively; Appendix A).

### 3.8. PET Imaging

Overall, the BxPC-3 tumors were clearly visualized by PET imaging with both peptides at all time points (Figure 6); the PET imaging also showed that [^64^Cu]**2** provided the clearest images based on its superior tumor-to-background ratios. Most notably, as previously discussed for the biodistribution data, the PET images for [^64^Cu]**1** had much higher kidney accumulation and higher levels of radiation in the liver, indicative of possible in vivo instability of [^64^Cu]**1**, which had shown substantially higher degradation in mouse serum compared to [^64^Cu]**2**.

## 4. Discussion

Cancer remains a leading cause of death globally [48,49]. Many cancers exhibit high expression of the cell surface receptor integrin α_v_β_6_, and expression levels correlate with poor prognosis and reduced progression-free and overall survival [31,32,38]. Therefore, integrin α_v_β_6_ has been identified as an important target both for imaging and treatment [50,51]. Receptor targeted delivery of radiopharmaceuticals is an important part of new approaches for improved cancer detection and therapy [48]. Peptides are attractive radiopharmaceuticals for both detection and treatment, because they are readily synthesized and can be chemically modified to optimize pharmacokinetics and metabolic stability. The addition of albumin binding moieties (ABMs) to numerous radiopharmaceuticals has demonstrated increased circulation time, reduced kidney uptake, and substantially increased tumor accumulation [18,52,53]. However, differences in the chemical structures of the ABM have been found at times to significantly affect the biodistribution, which ultimately determines target uptake, therapeutic efficacy, and off-target toxicity [52,54,55,56]. Thus, evaluation of different ABMs is important for optimal radiopharmaceutical performance towards the development of an α_v_β_6_-targeted radiotherapeutic agent. Our laboratory continues to develop integrin α_v_β_6_-targeting radiopharmaceuticals, including optimization of the core peptide structure [30] via PEGylation [14], and most recently the addition of an 4-(*p*-iodophenyl)butyryl (IP) ABM, which has demonstrated improved accumulation in tumors for both the [^18^F]AlF NOTA and [^64^Cu]Cu DOTA radiolabeled IP-ABM-α_v_β_6_-BP compared to the parent non-ABM α_v_β_6_-BP [42,44]. To further evaluate the choice of preferred ABM for α_v_β_6_-BP, the comparison of the IP-ABM with another prominent ABM, the Evans blue fragment (EB-ABM), was explored. The synthesis of both α_v_β_6_-BP peptides containing different ABMs, [^64^Cu]**1** or [^64^Cu]**2** (Figure 1), was done efficiently using a solid-phase approach, which allowed installation of the respective ABM-peptide from the same batch of peptidyl-resin by first coupling an orthogonally protected lysine allowing for the attachment of the DOTA-chelator at the *N*-terminus and either the EB-ABM **8** or the IP-ABM at the sidechain. The IP-ABM included an aspartate (D) residue as it is reported to result in better tumor retention [28]. After removal from the resin and purification, both DOTA-ABM-α_v_β_6_-BP peptides (**1** and **2**) were efficiently radiolabeled with copper-64 to yield [^64^Cu]**1** and [^64^Cu]**2** in high radiochemical purity >97%.

The ABM containing peptides [^64^Cu]**1** and [^64^Cu]**2** both demonstrated high tumor uptake at 4 h p.i., over 5% and 7.5% ID/g, respectively; representing a greater than 3-to-4.5-fold increase, respectively, from the non-ABM bearing [^64^Cu]Cu DOTA-α_v_β_6_-BP (1.61 ± 0.70% ID/g) [44]. The improvement in tumor accumulation was greater for the IP-ABM peptide [^64^Cu]**2** than for the EB-ABM peptide [^64^Cu]**1**, and was in concordance with the cell binding to both DX3puroβ6 and BxPC-3 cells (Figure 2). Furthermore, the prolonged tumor uptake and retention (Figure 4A) were maintained for 72 h, and, in conjunction with rapid renal clearance, provided a high tumor-to-background ratio (Figure 5) and high contrast PET-images (Figure 6). Since the only difference between [^64^Cu]**1** and [^64^Cu]**2** is the ABM, and [^64^Cu]**2** showed significantly higher stability in serum compared to [^64^Cu]**1** (Figure 3), the observed differences in the tumor-to-background ratios could be attributed to the improved stability. This study adds to the growing number of literature reports describing improved tumor uptake following the incorporation of ABMs [4,7,11]. For example, the small molecule PSMA-617, a radiopharmaceutical targeting the prostate specific membrane antigen (PSMA), exhibited approximately a fivefold increase in tumor accumulation with the addition of an EB-ABM at 4 h and a twofold increase for the IP-ABM modified PSMA-617, compared to the unmodified (non-ABM bearing) PSMA-617; furthermore, the EB-ABM PSMA-617 maintained tumor accumulation over time (65.6–77.3% ID/g from 4 h to 48 h) [55]. In another study with PSMA-617, the addition of the IP-ABM also resulted in twofold higher accumulation in tumor tissue as compared to the non-ABM containing PSMA-617 agent (non-ABM PSMA-617: 38% ID/g vs. IP-ABM PSMA-617: 75.7% ID/g at 24 h) [28,57]. Other small molecule PSMA agents modified with ABMs have also shown improvements in tumor accumulation, with the EB-ABM MCG PSMA agent having around a fourfold increase in tumor accumulation (MCG non-ABM: 10.9% ID/g vs. MCG-ABM: 40.4% ID/g at 24 h) [53] and an IP-ABM PSMA agent CTT1403 exhibiting >18-fold improvement in tumor accumulation (CTT1401 non-ABM: 2.2% ID/g vs. CTT1403-ABM: 40% ID/g at 24 h [54]. The addition of ABMs to other small molecule radiopharmaceuticals has also been shown to improve tumor accumulation with the small molecule radioligand folic acid modified with the IP-ABM having a threefold increase in tumor accumulation (ABM: 19.5% ID/g vs. non-ABM: 7% ID/g at 24 h, p.i.) with a considerably lower kidney accumulation (ABM: 28% ID/g vs. non-ABM: 70% ID/g at 4 h) [52,58].

Aside from small molecule radiopharmaceuticals, substantial benefits from the addition of ABMs to peptide radiopharmaceuticals have been shown; for example, the large peptide exendin-4 (39 amino acids), which targets the glucagon-like peptide 1 (GLP-1) receptor, when modified with the IP-ABM, demonstrated an improved stability and a twofold increase in tumor accumulation at 4 h, along with reduced kidney retention by more than half [7,59]. The small five amino acid integrin α_v_β_3_ targeting cyclic peptide (cRGDfK) modified with EB-ABM and radiolabeled as [^64^Cu]Cu NOTA-EB-cRGDfK displayed a >16-fold improvement (vs. [^64^Cu]Cu NOTA-cRGDfK) in tumor accumulation in a U87MG glioblastoma tumor model (with ABM: 16.6% ID/g vs. non-ABM: <1.1% ID/g), but only had about a fivefold improvement in MDA-MB-435 melanoma and HT29 colorectal adenocarcinoma models [18]. The somatostatin receptor targeting peptide octreotide (TATE), which is eight amino acids in size, has seen some of the greatest improvements in tumor accumulation upon modification with an ABM. For example, the EB-ABM modified [^177^Lu]Lu DOTA-EB-TATE provided a greater than eightfold increase in the tumor accumulation at 24 h (with ABM: 78.8% ID/g vs. non-ABM: 9.3% ID/g, respectively) [60] and the [^86^Y]Y DOTA-EB-TATE showed a larger enhancement with a between 30- and 60-fold increase in tumor accumulation compared to [^86^Y]Y DOTA-TATE, depending on the tumor model [6]. These studies paved the way for clinical trials where [^177^Lu]Lu DOTA-EB-TATE showed an extended circulation which led to a 7.9-fold increase in tumor dose delivery [61]. Overall, these studies illustrate the potential benefits of including an ABM on targeted peptide receptor radionuclide therapy (PRRT).

The addition of either EB-ABM or the IP-ABM on the α_v_β_6_-BP did significantly increase tumor accumulation (three-to-fivefold from the non-ABM-α_v_β_6_-BP) and the overall clearance properties of the ABM-modified α_v_β_6_-BP peptides [^64^Cu]**1** and [^64^Cu]**2** were similar with predominantly renal excretion. The organ with the highest accumulation was the kidneys, with the initial kidney uptake of the EB-ABM peptide [^64^Cu]**1** having more than double that of the IP-ABM peptide [^64^Cu]**2** (4 h: 75.5 ± 7.3% ID/g vs. 33.6 ± 5.4% ID/g, *p* = 0.0013), with both dropping to approximately one third of their initial value at 72 h p.i. (20.0 ± 6.9% ID/g and 11.4 ± 1.0% ID/g, respectively, *p* = 0.10, Figure 4). The introduction of the IP-ABM to the α_v_β_6_-BP significantly reduced kidney accumulation, which we hypothesize is due to the higher stability of the IP-ABM [^64^Cu]**2** over the EB-ABM [^64^Cu]**1**. These data are promising and indicate that renal toxicity would be less of a concern for PRRT of α_v_β_6_-BP agents using the IP-ABM. The observed effects of the different ABMs on kidney uptake and retention are comparable to other radiopharmaceutical ABM-adducts, for example, the ABM modified peptide [^177^Lu]Lu DOTA-TATE showed that the IP-ABM-analogue also provided lower kidney accumulation that was more rapidly cleared (dropping from ~20% ID/g at 4 h to ~5% ID/g at 72 h) compared to the EB-ABM-analogue (~30% ID/g at 4 h to ~15% ID/g at 72 h) [29,60]. This similar kidney accumulation and retention trend was also observed with the small molecule PSMA-617 agent, where the EB-PSMA-617 had considerably higher kidney accumulation and retention compared to the IP-PSMA-617, which had rapid kidney clearance (EB-PSMA-617: >20% ID/g at 4 h, which remained at 48 h vs. IP-ABM-PSMA-617: ~10% ID/g at 4 h dropping to <5% ID/g at 48 h) [55]. Both [^64^Cu]**1** and [^64^Cu]**2** also displayed some secondary clearance through the gastrointestinal (GI) tract and excretion of radioactivity in the feces (Appendix A). The IP-ABM modified peptide [^64^Cu]**2** had higher GI accumulation, with the highest uptake in the stomach of 18.1 ± 2.9% ID/g at 4 h, though, gratifyingly, both peptide’s GI accumulation dropped down to less than one-fifth of their respective original value (≤3.2% ID/g at 72 h, Figure 4).

The non-α_v_β_6_-targeting ABM controls [^64^Cu]**3** and [^64^Cu]**4** were used to evaluate non-specific uptake and demonstrate that the enhanced tumor accumulation of [^64^Cu]**1** or [^64^Cu]**2** resulted from integrin α_v_β_6_ receptor mediated uptake, as opposed to the enhanced permeability and retention (EPR) effect. As expected, [^64^Cu]**3** and [^64^Cu]**4** largely remained in the blood, thus mostly acting as blood pool imaging agents with high blood accumulation of 39.0% ID/g and 9.5% ID/g, respectively, at 4 h (Appendix A) and mirrored other similar non-targeted ABMs, such as the EB-ABM compound [^64^Cu]Cu NOTA-EB (NEB, ~15% ID/g at 4 h, dropping to ~10% ID/g at 1 d) [16,23]. Compared to the ABM peptides [^64^Cu]**1** and [^64^Cu]**2**, accumulation of [^64^Cu]**3** and [^64^Cu]**4** generally increased in organs with high blood flow (viz. heart, liver, and lungs; Figure 5A) but was lower in the kidneys (though the EB compound was still higher than the IP compound with 18.6% ID/g and 4.3% ID/g, respectively, at 4 h; Figure 5A), highlighting the effect of both the properties of the ABM as well as the targeting peptide moiety on kidney uptake. Both non-targeted [^64^Cu]**3** and [^64^Cu]**4**, due to their much higher blood accumulation (>9–39-fold higher than [^64^Cu]**1** and [^64^Cu]**2**) and longer blood residence time, provided much higher tumor accumulation at 4 h than the two peptides [^64^Cu]**1** and [^64^Cu]**2** (Figure 5A). However, the non-targeted [^64^Cu]**3** and [^64^Cu]**4** showed minimal binding (<4.3%) in cell binding studies to both the α_v_β_6_-expressing and α_v_β_6_-null cells (Appendix A), thus their higher tumor accumulation compared to [^64^Cu]**1** and [^64^Cu]**2** was attributed to the EPR effect (which, together with the long circulation, resulted in the expectedly low tumor-to-blood ratios of <0.9/1 (Figure 5B, Appendix A). By comparison, [^64^Cu]**1** and [^64^Cu]**2** showed high and α_v_β_6_-dependent cell binding (>30–60% binding; ~20:1 for DX3puroβ6 (+)/DX3puro (−) cells), and in vivo tumor uptake was efficiently blocked by the pre-administration of metal free **1** and **2**, respectively, supporting integrin α_v_β_6_-dependent tumor accumulation (Appendix A). Taken together, the tumor uptake observed for the integrin α_v_β_6_-binding peptides [^64^Cu]**1** and [^64^Cu]**2** was attributed to specific targeting of the integrin α_v_β_6_ receptor. Both ABM modified α_v_β_6_-BP peptides had improved pharmacokinetic profiles from the parent peptide and overall [^64^Cu]**2** demonstrated a more favorable biodistribution. Tumor retention of [^64^Cu]**1** and [^64^Cu]**2** was good over the three day study period, with each retaining about two-thirds of the original (4 h) uptake at 72 h p.i. The PET image quality improved, most notably for [^64^Cu]**2** over time after the initial uptake period (i.e., after 24 h p.i.) as a result of faster washout from non-target tissues (Figure 6). The high absolute tumor uptake of [^64^Cu]**2**, its efficient binding and internalization to α_v_β_6_-expressing cells (Figure 2), and its better serum stability (Figure 3) demonstrate the potential of using the [^64^Cu]**2** as an integrin α_v_β_6_-targeted peptide receptor radionuclide therapy (PRRT) agent where the copper-64 is replaced by a therapeutic radioisotope such as lutetium-177.

## 5. Conclusions

The effect of Evans blue (EB) and 4-(*p*-iodophenyl)butyryl (IP)-based albumin binding moieties (ABMs) on the pharmacokinetics of α_v_β_6_-BP, a peptide targeting the cancer-associated cell surface receptor integrin α_v_β_6_ was investigated. The albumin binding moieties on α_v_β_6_-BP did not interfere with integrin α_v_β_6_ affinity or selectivity in vitro. In vivo in a BxPC-3 pancreatic tumor xenograft mouse model, the IP-ABM-modified α_v_β_6_-BP [^64^Cu]**2** had a considerably more favorable pharmacokinetic profile compared to the EB-ABM-modified α_v_β_6_-BP [^64^Cu]**1**, with higher tumor uptake, reduced kidney and liver uptake, and improved tumor-to-background ratios that led to a clearer tumor visualization by PET imaging. Furthermore, the IP-ABM-modified α_v_β_6_-BP [^64^Cu]**2** had superior serum stability, making it a lead candidate for future integrin α_v_β_6_-targeted imaging and therapy studies.

## Data Availability

Additional data supporting the reported results can be found in the Appendix A.

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
