# Peer review of "A Comparison of Evans Blue and 4-(p-Iodophenyl)butyryl Albumin Binding Moieties on an Integrin αvβ6 Binding Peptide"

_pharmaceutics, 2022, doi:10.3390/pharmaceutics14040745_

Round 1

Reviewer 1 Report

In this work, the authors conjugate an Evans Blue-based albumin binding moiety (ABM) and iodophenyl butyryl ABM to an avB6 peptide and compare uptake of the Cu-64 labeled compound. The IP-ABM compound outperforms the EB-ABM in both tumor uptake and TBR. This direct comparison could be useful to the scientific community. However, there are several questions around the mechanism behind the results that would improve the quality of the work and strength of conclusions.

There are several factors that could explain the higher uptake and TBR of compound 2 including plasma protein binding, cellular specificity, and serum stability. Was the plasma clearance of these compounds measured? With such high renal clearance, the 4 hr time point can’t detect much difference in the AUC. The plasma clearance could help shed light on the mechanism (e.g. slower clearance generally results in higher tumor uptake).

The general trend for many agents is slower clearance results in higher tumor uptake (%ID/g) but lower TBR, whereas faster clearance gives higher TBR with lower tumor uptake. However, compound 2 shows higher uptake and TBR. Is this attributed primarily to the plasma stability? It’s hard to say without an estimate of the plasma clearance (likely biexponential).

The lower renal uptake of IP is a promising result compared to EB-ABM, which may be one of the most interesting conclusions from the paper. Is there any speculation on the reason for lower renal uptake with the IP-ABM versus the EB-ABM? Is it due to slower renal clearance for IP-ABM, higher clearance via other mechanisms (liver/GI), reduced uptake in the proximal tubule, etc? Or is it based on stability where degraded compound 1 is cleared renally?

Line 564 – is this for therapeutics or imaging agents? The goals of these are often different even if parts are shared. Imaging is typically focused on contrast (e.g. TBR) with less emphasis on total uptake as long as the signal strength is sufficient. Therapy often focuses on target uptake (required for a therapeutic response). While the authors could (and likely are) examining both applications, some clarity would help, since the abstract discusses imaging but the manuscript mentions PRRT several times.

Line 399 – typo, concious

Line 560 – typo, albumn

Line 628 – typo, clinical trails

Author Response

Reviewer 1

In this work, the authors conjugate an Evans Blue-based albumin binding moiety (ABM) and iodophenyl butyryl ABM to an avB6 peptide and compare uptake of the Cu-64 labeled compound. The IP-ABM compound outperforms the EB-ABM in both tumor uptake and TBR. This direct comparison could be useful to the scientific community. However, there are several questions around the mechanism behind the results that would improve the quality of the work and strength of conclusions.

Reviewer 1. Comment 1. 

There are several factors that could explain the higher uptake and TBR of compound 2 including plasma protein binding, cellular specificity, and serum stability. Was the plasma clearance of these compounds measured? With such high renal clearance, the 4 hr time point can’t detect much difference in the AUC. The plasma clearance could help shed light on the mechanism (e.g. slower clearance generally results in higher tumor uptake).

Authors’ Response to reviewer 1, comment 1.

We would like to thank reviewer 1 for the review and insightful thoughts on the TBR and AUC in the blood and agree that several factors might explain the higher uptake and TBR. Based on our earlier work with this peptide (Hausner 2020, Ganguly 2021), in this manuscript we measured in vivo clearance data starting at 4 hours and showed the ABMs’ effects on slowing plasma clearance and providing prolonged tumor uptake and retention (as depicted in Figure 4, A the tumor accumulation is maintained to a high level 72 hours later). Given that the only difference between [64Cu]1 and [64Cu]2 is the albumin binding moiety (ABM), and that [64Cu]2 showed significantly higher stability in serum compared to [64Cu]1 (Figure 3, 58% intact [64Cu]1 vs 83% intact [64Cu]2, at 4 h), we believe that stability plays a major role in the observed differences in tumor-to-background ratios.

For clarification we have now updated the discussion to include the following sentences placed in the 2nd paragraph, line 588: “Furthermore, the prolonged tumor uptake and retention (Figure 4, A.) were maintained for 72 h, and, in conjunction with rapid renal clearance, provided a high tumor-to-background ratio (Figure 5) and high contrast PET images (Figure 6). Since, the only difference between [64Cu]1 and [64Cu]2 is the ABM, and [64Cu]2 showed significantly higher stability in serum compared to [64Cu]1 (Figure 3), the observed differences in the tumor-to-background ratios could be attributed to the improved stability.  

Hausner, S. H.; et al. The effects of an albumin binding moiety on the targeting and pharmacokinetics of an integrin αvβ6-selective peptide labeled with aluminum [18F]fluoride. Mol. Imaging Biol. 2020, 22, 1543-1552.

Ganguly, T.; et al. Evaluation of copper-64-labeled αvβ6‑targeting peptides: Addition of an albumin binding moiety to improve pharmacokinetics. Mol. Pharmaceutics. 2021, 18, 4437-4447.

Reviewer 1. Comment 2. 

The general trend for many agents is slower clearance results in higher tumor uptake (%ID/g) but lower TBR, whereas faster clearance gives higher TBR with lower tumor uptake. However, compound 2 shows higher uptake and TBR. Is this attributed primarily to the plasma stability? It’s hard to say without an estimate of the plasma clearance (likely biexponential).

Authors’ Response to reviewer 1, comment 2.

We thank the reviewer and agree that generally, slower clearance results in higher tumor uptake but lower TBR, whereas faster clearance gives higher TBR with lower tumor uptake. As mentioned above, multiple factors might influence the tumor uptake and TBR for example stability. While we cannot say with certainty that the improved performance of [64Cu]2 vs [64Cu]1 is primarily attributable to difference in plasma stability, as stated  above the only difference between [64Cu]1 and [64Cu]2 is the albumin binding moiety (ABM), and as [64Cu]2 showed significantly higher stability in serum we speculate that this is the case. In addition, prior literature describing other ABM-modified radiopharmaceuticals also shows a trend that an EB-ABM modification has higher kidney uptake compared to the IP-ABM modification (Rousseau 2018, Wang 2018, Bandara 2018). We agree that a more in-depth plasma clearance study would indeed be very interesting for EB and IP based ABMs, particularly in light of their increasing popularity, but we feel that would be beyond the scope of the present study as our goal was to directly compare the two ABMs when conjugated to the integrin avb6-binding peptide (avb6-BP).   

Rousseau, E.; et al. Effects of adding an albumin binder chain on [177Lu]Lu-DOTATATE. Nucl. Med. Biol. 2018, 66, 10-17.

Wang, Z.; et al. Single low-dose injection of Evans blue modified PSMA-617 radioligand therapy eliminates prostate-specific membrane antigen positive tumors. Bioconjugate Chem. 2018, 29, 3213-3221.

Bandara, N.; et al. Novel structural modification based on Evans blue dye to improve pharmacokinetics of a somastostatin-receptor-based theranostic agent. Bioconjugate Chem. 2018, 29, 2448-2454.

Reviewer 1. Comment 3. 

The lower renal uptake of IP is a promising result compared to EB-ABM, which may be one of the most interesting conclusions from the paper. Is there any speculation on the reason for lower renal uptake with the IP-ABM versus the EB-ABM? Is it due to slower renal clearance for IP-ABM, higher clearance via other mechanisms (liver/GI), reduced uptake in the proximal tubule, etc? Or is it based on stability where degraded compound 1 is cleared renally?

Authors’ Response to reviewer 1, comment 3.

We were equally, and pleasantly, encouraged by the lower kidney uptake observed for [64Cu]2 vs [64Cu]1, particularly since they are the organ with the highest uptake. The lower renal uptake of the IP-ABM peptide [64Cu]2 versus the EB-ABM peptide [64Cu]1 could indeed be due to all of the reasons stated by this reviewer. We hypothesize that for avb6-BP the IP-ABM peptide’s lower renal uptake is due to the increased stability of the IP-ABM [64Cu]2, but this is only speculation on the observed outcome. This observation, though, is consistent with the above mentioned reports on kidney uptake (Rousseau 2018, Wang 2018, Bandara 2018) comparing IP-ABM and EB-ABM radiopharmaceuticals.

To further highlight this promising result of the lower kidney uptake with IP-ABM [64Cu]2, we have modified the discussion and included a sentence after line 640 in the 4th paragraph stating: “The introduction of the IP-ABM to the avb6-BP significantly reduced kidney accumulation, which we hypothesize is due to the higher stability of the IP-ABM [64Cu]2 over the EB-ABM [64Cu]1.” 

Reviewer 1. Comment 4. 

Line 564 – is this for therapeutics or imaging agents? The goals of these are often different even if parts are shared. Imaging is typically focused on contrast (e.g. TBR) with less emphasis on total uptake as long as the signal strength is sufficient. Therapy often focuses on target uptake (required for a therapeutic response). While the authors could (and likely are) examining both applications, some clarity would help, since the abstract discusses imaging but the manuscript mentions PRRT several times.

Authors’ Response to reviewer 1, comment 4.

We agree with the reviewer that, while parts are shared, the goals are often different between an imaging agent and a therapeutic agent, as we have also noted in the Introduction, lines 45-48, “However, some of the properties that are desirable for a diagnostic agent can hamper the translation to a therapeutic, which relies on a prolonged circulation for high and persistent uptake in the targeted tissue.”
The sentence in line 564 is referring to the effects of ABMs on an agent used as a therapeutic. To clarify this we have updated lines 565-566 “towards the development of an avb6-targeted radiotherapeutic agent.” to now read “Thus, evaluation of different ABMs is important for optimal radiopharmaceutical performance towards the development of an avb6-targeted radiotherapeutic agent.

Reviewer 1. Comment 5. 

Line 399 – typo, conscious

Line 560 – typo, albumn

Line 628 – typo, clinical trails

Authors’ Response to reviewer 1, comment 5.

Thank you for bringing these typos to our attention. They have been corrected:

Line 399 - corrected to “conscious”

Line 560 - corrected to “albumin”

Line 628 - corrected to “clinical trials”

Reviewer 2 Report

The author describe the design and the synthesis of a combination of an integrin binding peptide with Evans Blue (EB) and 4-iodophenylbutyryl (IP)  together with a chelator for radiolabeling studies. They describe the synthetic steps for obtaining the probes and analyze extensively the distribution of both novel conjugates with comparison of the native conjugates. In addition they also performed serum stability studies and albumin binding experiments. It is an interesting study with proper compilation of recent literature in this specific field and comparing the different strategies of increasing tumor uptake of therapeutic agents. It is very well written and clear, and the SI also gives additional information relevant to the publication.

I have some suggestion, regarding the uptake of the compounds when lacking the αvβ6-BP. I believe it needs further figures in the main text and elaborating on this comparison. To complete the manuscript:

[64Cu]3 had a very high Tumor uptake that needs additional clarification and statements in the abstract, conclusions and text, since just at absolute values it is higher that [64Cu]1 and [64Cu]2 (table at S25). When adding the αvβ6-BP, the increased tumor concentration is due to the lower penetration to other organs right? This should be stated in every part of the paper to be clear. Also, this introduction increased kidney uptake significantly and was addressed in the text. I believe the last figure from the SI needs to be in the main text to be further explained. Also I believe a combination of figures at S20 and S21 with figure at S25 would be great to have the visual comparison of 4h distribution of the 4 compounds. I would also add this figure to the main text.

Also, the manuscript is centered in 1,2,3,4 comparison but would highly improve if some comparison would be introduced without the EB or IP or the quantitative measurements mentioned in the text, to really see the improvement of assing these scaffolds to the conjugates.

Finally, there are very small chromatograms in SI, please make Chromatogram of DOTA-EB-avb6-BP bigger as it is very difficult to see properly in the SI, same as RP-HPLC of DOTA-EB-ABM 3, 4, 6, 8

Author Response

Reviewer 2

The author describe the design and the synthesis of a combination of an integrin binding peptide with Evans Blue (EB) and 4-iodophenylbutyryl (IP)  together with a chelator for radiolabeling studies. They describe the synthetic steps for obtaining the probes and analyze extensively the distribution of both novel conjugates with comparison of the native conjugates. In addition they also performed serum stability studies and albumin binding experiments. It is an interesting study with proper compilation of recent literature in this specific field and comparing the different strategies of increasing tumor uptake of therapeutic agents. It is very well written and clear, and the SI also gives additional information relevant to the publication.

Reviewer 2. Comment 1. 

I have some suggestion, regarding the uptake of the compounds when lacking the αvβ6-BP. I believe it needs further figures in the main text and elaborating on this comparison. To complete the manuscript:

[64Cu]3 had a very high Tumor uptake that needs additional clarification and statements in the abstract, conclusions and text, since just at absolute values it is higher that [64Cu]1 and [64Cu]2 (Table at S25). When adding the αvβ6-BP, the increased tumor concentration is due to the lower penetration to other organs right? This should be stated in every part of the paper to be clear. Also, this introduction increased kidney uptake significantly and was addressed in the text. I believe the last figure from the SI needs to be in the main text to be further explained. Also I believe a combination of figures at S20 and S21 with figure at S25 would be great to have the visual comparison of 4h distribution of the 4 compounds. I would also add this figure to the main text.

Authors’ Response to reviewer 2, comment 1.

We thank the reviewer for the comments and thoughtful suggestions. In response, the following additions and changes have been made to the figures in the text: Figure 5 is now a combination of S20, S21 and S25 showing the biodistribution of [64Cu]1-4 at 4 hours in select tissues (Part A.) and  the last figure from the SI comparing the tumor-to-organ ratios of [64Cu]1-4 at 4 hours (Part B.). The original Figure 5 showing the PET images is now Figure 6.       

Regarding the organ and tumor uptake for ABM vs ABM-peptide, we agree that this can be viewed as being a result of adding the ABM to the peptide, where the ABM by itself shows generally higher organ uptake due to the high % ID/g in the blood pool. To address this, we have updated the discussion to further emphasize the point that the non-targeted ABM moieties have higher tumor uptake; and to incorporate the new Figure 5, and the following modifications to the last paragraph starting at line 669, which now reads:  “Both non-targeted [64Cu]3 and [64Cu]4, due to their much higher blood accumulation (>9-39-fold higher than [64Cu]1 and [64Cu]2) and longer blood residence time, provided much higher tumor accumulation at 4 h than the two peptides [64Cu]1 and [64Cu]2 (Figure 5, A.). However, the non-targeted [64Cu]3 and [64Cu]4 showed minimal binding (<4.3%) in cell binding studies to both the avb6-expressing and avb6-null cells (S24), thus their higher tumor accumulation compared to [64Cu]1 and [64Cu]2 was attributed to the EPR effect (which, together with the long circulation, resulted in the expectedly low tumor-to-blood ratios of <0.9/1 (Figure 5, B.).”

Reviewer 2. Comment 2. 

Also, the manuscript is centered in 1,2,3,4 comparison but would highly improve if some comparison would be introduced without the EB or IP or the quantitative measurements mentioned in the text, to really see the improvement of assing these scaffolds to the conjugates.

Authors’ Response to reviewer 2, comment 2.

We agree that comparing these data to the peptide without the ABM is important and would like to highlight that the initial comparison without the ABM modifications to the avb6-BP peptide was previously described in (Ganguly 2021) and as such we had felt the inclusion of the results previously described was not appropriate.

However, to facilitate the comparison we have now added the following key data from our prior study to the results section 3.5 starting at line 470 to now read:  “The ABM modifications of [64Cu]1 and [64Cu]2 increased the serum albumin affinity as the [64Cu]Cu DOTA-avb6-BP without an ABM modification showed <29% binding to either serum albumin [44].”

In addition, the results section 3.6 starting at line 487 now contains the new sentence: “The ABM modifications increased tumor accumulation by >3-to-4.5 fold compared to the [64Cu]Cu DOTA-avb6-BP without an ABM, which had only 1.61±0.70% ID/g at 4 h in the same BxPC-3 tumor model [44].”

And the results section 3.6 starting at line 493 now contains the new sentence: “Kidney accumulation for the ABM containing peptides [64Cu]1 and [64Cu]2 was initially higher than for the parent non-ABM containing [64Cu]Cu DOTA-avb6-BP (20.37±1.67% ID/g at 4 h to 6.81±1.36% ID/g at 48 h) [44].”

Ganguly, T.; et al. Evaluation of copper-64-labeled αvβ6‑targeting peptides: Addition of an albumin binding moiety to improve pharmacokinetics. Mol. Pharmaceutics. 2021, 18, 4437-4447. (Reference 44)

Reviewer 2. Comment 3. 

Finally, there are very small chromatograms in SI, please make Chromatogram of DOTA-EB-avb6-BP bigger as it is very difficult to see properly in the SI, same as RP-HPLC of DOTA-EB-ABM 3, 4, 6, 8

Authors’ Response to reviewer 2, comment 3.

Thank you for bringing this to our attention. The size of the HPLC chromatograms in the SI has all been increased.

Round 2

Reviewer 2 Report

Thank you for the revised version, I think the points raised have been fullfilled.